# Neural correlates of the deployment of spatial attention, and their modulation by repetitive movements

**Cameron Smith, Daniel H. Baker** ID *

Department of Psychology, University of York, York, United Kingdom

* daniel.baker@york.ac.uk

## Abstract

The deployment of spatial attention generates distinct neural signatures that can be detected at the scalp. Here, we use multivariate pattern analysis of EEG data to decode the deployment of spatial attention, and ask if this is modulated by repetitive movements. 'Stimming' movements (also known as repetitive stereotypies), are widely reported in autism, but also present in some neurotypical individuals. Stimming has historically been viewed as a problematic behaviour, but many individuals claim that stimming benefits attention. We first validated our paradigm (a Posner-style cueing design), demonstrating above-chance classification of cue direction from around 300 ms post-cue onset. We then investigated whether stimming modulates decoding accuracy and task performance. Our results, consisting of data primarily from neurotypical participants, do not suggest that stimming has a negative impact on an individual's ability to attend, unless the individual does not typically engage in stimming behaviours. This suggests interventions aiming to reduce stimming behaviours are not necessarily warranted and highlights the need for further research into the potential benefits of stimming specifically within the autistic population. Future research might also consider the potential overlap between autistic stimming and the fidgeting behaviours which are characteristic of ADHD, to help understand the significant overlaps between the characteristics of the two conditions.

## Introduction

The ability to orient and control the deployment of our attention to spatial locations or features of our environment is essential to our ability to perform tasks, focusing on relevant stimuli and disregarding those that are unnecessary for the task at hand [1–3]. A classic technique for studying the deployment of attention in space is the Posner cueing paradigm, which demonstrates that it is possible for attention to be deployed to spatial locations independently of eye movements [4].

Individuals with neurodevelopmental disorders such as attention deficit hyperactivity disorder (ADHD) and autism present with deficits in attention [5]. It has been suggested that repetitive stereotypies, or 'stimming' behaviours, are linked to these deficits in attention – in the few studies where they have been asked, individuals who stim report that it benefits

**Data availability statement:** Raw data, processed data, and analysis scripts are freely available through the project repository at: http://doi.org/10.17605/OSF.IO/YUHZ4.

**Funding:** CS was supported by a Bradshaw-Eagle Undergraduate Research Scholarship from the Applied Vision Association (https://www.theava.net/). DHB was supported by BBSRC grant BB/V007580/1 (https://www.ukri.org/councils/bbsrc/). There was no additional external funding received for this study. The funders played no role in the study design, data collection and analysis, decision to publish, or preparation of the manuscript.

their ability to focus, and that the behaviours aid in emotional regulation [6–10]. It has been proposed that an improved ability to focus through stimming may have a neural basis [11]. Building on these theories and research, we tested the effect of stimming on performance in a Posner cueing task.

The Posner cueing paradigm [4] involves the participant attending to a central fixation point, then receiving a cue (which may be exogenous, where attention is captured through bottom-up processes; or endogenous, where attention is controlled by top-down processes, such as attending selectively to cues of the correct colour [12]) indicating that a stimulus is about to appear either to the left or the right of the fixation point [13]. The assumption underlying the task is that if participants successfully orient their attention to the cued location, then their reaction times will be shorter and responses more accurate for congruent trials (where the stimulus appears at the cued location) than for incongruent trials (where the stimulus appears at the uncued location); this has been demonstrated extensively [13]. The reaction time difference was found by Posner [14] to be 25 ms faster for congruent trials and 40 ms slower for incongruent trials, compared to a neutral cue condition.

This paradigm has been combined with EEG to investigate the neural basis of spatial attention. For example, Landau et al. [15] investigated gamma-band activity during a Posner cueing task, comparing congruent and incongruent trials. Gamma range responses were associated with voluntary shifts in attention, e.g., attending to the left of the fixation point following a left-pointing arrow cue, but were absent when attention was involuntarily captured by the appearance of a stimulus in the uncued location on incongruent trials; this suggests that voluntary and involuntary processes of attention deployment are controlled by separate mechanisms. Thiery et al. [13] investigated two specific ERPs, the N2pc component (a negative response typically seen at around 200 ms post stimulus, in the posterior-contralateral region) and the SPCN component (sustained posterior contralateral negativity). They used multivariate pattern analysis and used the classifier accuracy to determine that these two components were able to predict the focus of participants' spatial attention. This was modulated by the varied distance of the stimulus from the central fixation (further being better decoded).

A combined EEG and TMS study by Capotosto et al. [16] demonstrated that the application of TMS to the right intraparietal sulcus (IPS) impairs target detection, particularly on incongruent trials. They additionally noted that the P3 response amplitude was affected by this; on incongruent trials its amplitude was significantly lower, and on congruent trials it was significantly higher. The IPS has been previously identified as an important region in attentional deployment by Vossel et al. [17] in an fMRI study identifying regions that are selectively active on congruent and incongruent trials. The right IPS and, additionally, the right inferior frontal gyrus were implicated in attentional processes related to both types of trial. Peelen et al. [18] concluded that both endogenous and exogenous orienting cues involved the same network in their fMRI study, implicating frontal and parietal regions (the premotor cortex, posterior parietal cortex, medial frontal cortex, and right inferior frontal cortex). Fitzgerald et al. [19] conducted an MRI study using the Posner cueing task and found that activation in the ventral attention network differed between individuals with autism and controls; individuals with autism also showed weaker functional connectivity in the dorsal attentional network compared to their neurotypical counterparts.

A number of developmental disorders involve deficits in attention, notably ADHD and autism. In the case of ADHD, attention deficits are a defining feature of the condition (American Psychological Association, 2013). For autism, attention deficits are not an explicit part of the diagnostic criteria (although restricted interests, which are included, are somewhat related), but both attentional deficits and strengths are widely reported in autistic individuals. They are typically more distractible, demonstrating 'underselective' attention, but also

demonstrate 'overselective' attention where they attend intensely to a more limited subset of stimuli. This 'overselective' attention has been suggested to be linked to restricted interests, which is also sometimes referred to as 'hyperfocus', where a task is the subject of intense focus for the individual (e.g., [20]). These differences appear to have a neural basis – they have been linked to unusually wide sulci in the parietal lobe [5]. Hyperfocus is also widely self-reported in ADHD, despite a lack of clinical research into the phenomenon [21].

These marked differences in attentional deployment seen in both conditions are worth considering in the framework of the transdiagnostic approach, which seeks to identify core deficiencies that result in the presentation of traits from multiple disorders, instead of approaching different disorders as entirely separate even within the same individual [22]. ADHD is the most common comorbidity of autism, and even without a dual diagnosis, 30-80% of individuals with autism present symptoms of ADHD, and 20-60% of children with ADHD exhibit autism-like traits [23]. Some shared deficiencies that have already been identified in the case of autism and ADHD include fine motor function and verbal fluency, alongside some structural differences in the brain [23].

Stimming behaviours are commonly seen in individuals with diagnoses of autism and/or ADHD, and involve repetitive movements, typically featuring the arms, hands, or entire body [24], for example hand flapping [6]. They are also observed in neurotypical individuals, but at much lower rates than in neurodiverse individuals [24]. The behaviours are typically defined as involuntary, but pleasant [6,7,24], and are associated with emotional states such as stress, boredom, concentration, or excitement [25]. They are often defined as purposeless (e.g., [26]) but self-reports from individuals with autism claim that stimming aids the regulation of their emotions [6,7] and concentration [8,10].

A type of vocal stimming called echolalia has been consistently demonstrated to have benefits for autistic children in the acquisition of language [27,28]. This is one of the few instances that stimming has been researched in terms of its potential function. Echolalic utterances, which typically involve repeating words spoken by another, have also been identified as having predominantly communicative functions [29]. This opens up an avenue to research potential benefits that other types of stimming may provide.

Current research into stimming is largely focussed on designing interventions to eliminate the behaviours, for example the use of weighted vests [30]. Taking this approach without first attempting to understand if there are benefits to the behaviour seems ill-advised. McCarty and Brumback [11], for example, proposed based on self-reports of the benefits of stimming (focus, coping with overwhelming sensory input, and relaxation) that regular motor movements may generate (or be a by-product of) rhythmic oscillations in the motor cortex, which entrain oscillations in the sensory cortex. This entrainment might improve information transfer in the sensory cortex, effectively normalising the atypical motor and sensory oscillations observed in the brains of autistic individuals both during tasks [31–33] and at rest [34,35].

There is heavy stigma around stimming behaviours, with many individuals reporting they have been explicitly instructed not to stim, despite attempts at repression causing them some discomfort [7]. Individuals with autism state that they believe stimming should not be stigmatised in this way, and that they should be allowed to act in the way that feels natural to them [6]. They typically report that stimming is a positive experience for them, and negative only when it is self-injurious or stigmatized [9,36], and there is a growing body of evidence to suggest stimming has a communicative function [36,37]. Research into potential benefits of stimming is important as a step towards dispelling this stigma, and discouraging interventions that aim to eliminate the behaviour.

This study involved two experiments. In the first experiment, participants completed a Posner cueing task while EEG recordings were taken; a pattern classifier then attempted to differentiate, using the participants' brain activity, whether they attended to the left vs. to the right following the cue onset. This is similar to the methodology used by Thiery et al. [13]: by measuring the timecourse of classifier accuracy, we are provided with an index of the deployment of spatial attention. The second experiment extended this paradigm by adding a stimming condition. The accuracy of the classifier was compared between conditions where the participants were permitted to stim during the trials, or instructed to keep still. In this case, we used decoding accuracy to assess whether stimming provided a benefit in terms of the deployment of spatial attention; if there was a benefit, the classifier accuracy would be higher, as the EEG signal would differ more between conditions due to the greater focus of attention in different spatial locations. For the purposes of our analyses, due to an insufficiently sized and heterogeneous clinical group, we grouped the participants based on whether or not they reported stimming in their everyday lives; as discussed previously, neurotypicals often engage in stimming [24], and 64% of our participants reported in the pre-experiment questionnaire that they typically engaged in stimming.

## Materials and methods

### Participants

41 participants took part in Experiment 1, but four produced very noisy data so were excluded, leaving a sample of 37 participants (20 male, 17 female). A total of 22 participants took part in Experiment 2 (8 male, 14 female). Participants in Experiment 2 were asked to report diagnoses of neurodevelopmental disorders – 17 had no diagnosis, 2 had a diagnosis of autism, and 3 had a diagnosis of ADHD. 15/22 participants further reported they typically engaged in stimming behaviours, while 7 reported that they did not - this information was collected through a single yes/no question asking if participants typically stim, due to a lack of available self-report stimming scales. Both experiments were granted ethical approval by the Ethics Committee of the Department of Psychology at the University of York. The identification numbers were 356 for Experiment 1, and 173 for Experiment 2. Participants provided written informed consent before data collection began. Data collection for Experiment 1 began on 25th January 2017, and ended on 10th March 2017. Data collection for Experiment 2 began on 14th July 2022, and ended on 14th September 2022.

### Apparatus and stimuli

Stimuli were displayed on a ViewPixx 3D display (VPixx Technologies Inc., Quebec, Canada) with a diagonal extent of 24 inches, a refresh rate of 120 Hz, and a resolution of 1920x1080 pixels. The screen was viewed by the participants from a distance of 57 cm, resulting in a spatial resolution of 36 pixels per degree of visual angle. The display was gamma corrected using a Minolta LS110 photometer to ensure a linear luminance response. Stimuli were generated and displayed using a Mac Pro computer running Matlab, with the experiment script making use of functions from Psychtoolbox 3 [38–40].

In both experiments, EEG recordings were acquired using an ANT Neuroscan 64-channel EEG system (ANT Neuro, Hengelo, Netherlands). Electrodes were mounted in a waveguard cap, arranged according to the 10/20 system, and were referenced to the whole head average. The ground electrode was located at position *AFz*, and EEG signals were recorded at 1kHz. In Experiment 2, participants' eye movements were also recorded using an EyeLink 1000

(SR Research Ltd., Ottawa, Canada) eye tracker, also sampling at 1kHz. Low-latency digital triggers were sent from the stimulus computer to the EEG system using an 8-bit parallel cable.

Target stimuli in both experiments were horizontal sine-wave gratings with a diameter of 4 degrees, and a spatial frequency of 2c/deg. The target appeared offset by 8 degrees to either the left or the right of the central fixation point. The base contrast of the target was 50%, and either the upper or lower half of the grating contained a contrast increment of a further 10%. Participants were asked to indicate whether the upper or lower half of the grating contained the increment using arrow keys on the computer keyboard. The fixation cross was $0.25 \times 0.25$ degrees, and was displayed throughout.

Each trial was preceded by a cue. In Experiment 1 the cue was a face with eyes pointing left or right. In Experiment 2 the cue was an arrow pointing left or right. The face cues used in Experiment 1 were the average of 22 female faces taken from the NimStim database [41] and were digitally altered so that the eyes were directed to the left or the right. The face had a width of 3 degrees and a height of 4 degrees and was displayed in the centre of the screen. The arrow cue used in Experiment 2 was 1.4 degrees long and was displayed in the centre of the screen.

## Procedure

Participants in both experiments completed four blocks of the Posner cueing task, with each block consisting of 200 trials. They were instructed to keep their eyes focused on the central fixation cross, and to attend to the gratings without shifting their gaze. During each trial, participants viewed a fixation cross, which remained on the screen throughout the trial (excluding times when the cue appeared on the screen). The cue stimulus appeared for 200 ms, followed by a randomly determined gap of 400–600 ms before the grating appeared on the screen. The target grating was then displayed for 200 ms. Participants were required to report whether the upper or lower half of the grating appeared higher in contrast; following their response, a randomly determined interval drawn from a normal distribution (average 1000 ms, with a standard deviation of ±200 ms) preceded the next appearance of the cue. This procedure is illustrated in Fig 1 for the arrow cue. Cue congruence was 75% in both experiments, meaning that each participant contributed a total of 600 congruent trials and 200 incongruent trials across the experiment.

In Experiment 2, participants were additionally instructed to stim, using a movement restricted to one hand only, in half of the blocks. This was to ensure minimal distortion of the EEG signal resulting from movement. The order in which participants completed stimming or non-stimming blocks was counterbalanced; half the participants were instructed to stim on blocks 1 and 3, while the other half were instructed to stim on blocks 2 and 4.

## Data analysis

Raw EEG data were converted into EEGlab .set format using Matlab and the EEGlab toolbox [42]. We then used MNE python [43] for the main analysis. Data were bandpass filtered between 0.2 and 30Hz, and then epoched using the event timestamps. We performed classification using a linear classifier with 10-fold cross-validation on the voltage data across all electrodes (excluding electrodes *M1* and *M2*), independently at each time point and for each participant. We then averaged classification accuracy across participants to generate the timecourses used in the main analysis. We calculate Bayes factors [44] to compare classification accuracy to chance performance (50% correct) and to make comparisons between conditions and groups, but also report frequentist tests for reference. We express Bayes factors in logarithmic units to account for extreme values ($log_{10}BF_{10}$), such that a Bayes factor of 3 is a

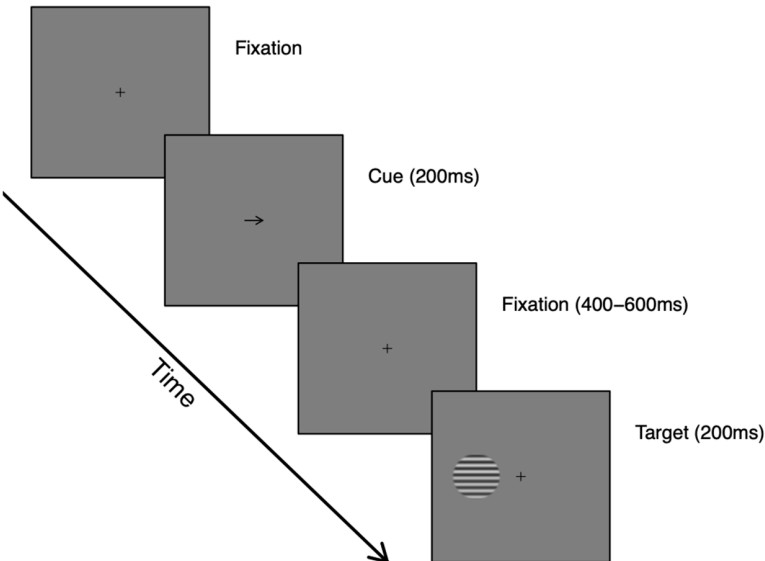

**Fig 1. Illustration of trial procedure for the second experiment.** The cue shown is an arrow, but in Experiment 1 the cue was a face with eyes pointing to either the left or the right. The face is not shown here for copyright reasons. Target gratings had a contrast of 50%, plus a 10% increment in either the upper or lower half. Note that images are not drawn to scale in this schematic to aid visibility - see text for actual dimensions.

$log_{10}BF_{10} = 0.48$, and constitutes convincing evidence for the alternative hypothesis, a Bayes factor of 10 is a $log_{10}BF_{10} = 1$, and constitutes strong evidence for the alternative hypothesis, and a Bayes factor of 30 is a $log_{10}BF_{10} = 1.48$, and constitutes overwhelming evidence for the alternative hypothesis.

## Results

### Experiment 1

We first analysed the behavioural responses from the experiment, comparing accuracy and reaction times between stimuli shown in the cued (i.e. a congruent target) and uncued (an incongruent target) location. There was a highly significant effect of cue congruency on both accuracy ($log_{10}BF_{10} = 1.77$, $t = 3.82$, $p < 0.001$, $d = 0.61$) and reaction time ($log_{10}BF_{10} = 6.08$, $t = -7.26$, $p < 0.001$, $d = 1.16$), as shown in Fig 2. Responses to targets in the cued location were both faster ($M_{diff} = 51$ ms) and more accurate ($M_{diff} = 3\%$) than for those in the uncued location, showing a classic Posner cueing effect.

We next explored the EEG data in response to both the cue presentation and the target presentation. There were typical evoked potentials in response to both types of stimulus, as summarised in Fig 3b, 3c, 3e, 3f. Patterns of voltage across the scalp showed similar lateralisation effects both for cues pointing to either the left or right, and for targets presented to either the left or right. Multivariate pattern analysis was able to decode the side of cue presentation from around 300 ms after cue onset (see Fig 3a), with a maximum accuracy of 69%, and a plateau of high performance from around 500–1000 ms. This relatively late onset of above-chance classification likely reflects the deployment of covert spatial attention to one side of the visual field, and is therefore a neural index of attentional focus. The pattern classifier could decode the side of target presentation with much greater accuracy (maximum of 89%), from around 100 ms after target onset (black curve in Fig 3d). Finally, we were able to decode cue

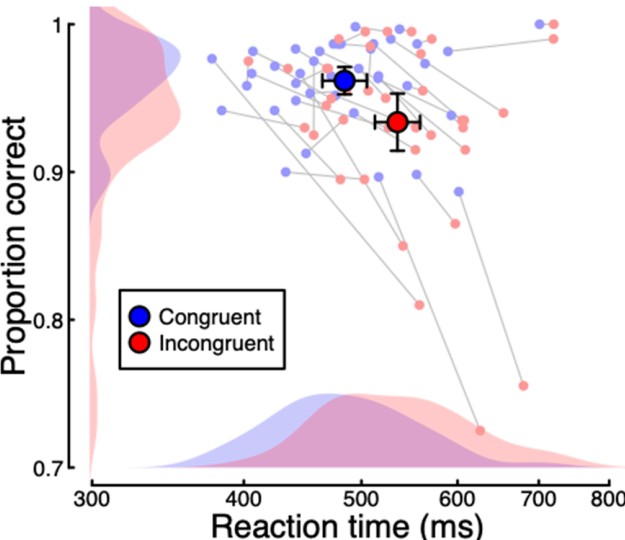

**Fig 2. Summary of reaction time and accuracy data for the cueing task in Experiment 1.** Small points show data for individual participants (N = 39), and larger points give the group means. Error bars indicate 95% confidence intervals.

congruency (i.e. whether the target presented was on the side indicated by the cue) with a maximum accuracy of 62%, and a similar timecourse to the decoding of target side (green curve in Fig 3d). Overall, these results indicate that our EEG data are sufficiently sensitive to provide an index of the deployment of spatial attention.

## Experiment 2

In Experiment 2, we replicated the basic design of Experiment 1, but this time with an added 'stimming' condition, and with participants divided into groups based on their self-reported stimming behaviour. The results are summarised in Fig 4. Pooling across all participants, for response accuracy there was no effect of cue congruency ($log_{10}BF_{10} = -0.65$, $F = 0.06$, $p = 0.803$, $\eta_G^2 = 0$), stimming condition ($log_{10}BF_{10} = -0.66$, $F = 0$, $p = 0.977$, $\eta_G^2 = 0$), nor an interaction between the two ($log_{10}BF_{10} = -0.48$, $F = 0.8$, $p = 0.383$, $\eta_G^2 = 0$). For reaction time, there were significant effects of cue congruency ($log_{10}BF_{10} = 3.1$, $F = 29.28$, $p < 0.001$, $\eta_G^2 = 0.07$) and stimming condition ($log_{10}BF_{10} = 1.08$, $F = 7.1$, $p = 0.01$, $\eta_G^2 = 0.04$), as well as an interaction between the two ($log_{10}BF_{10} = -0.34$, $F = 12.03$, $p = 0.002$, $\eta_G^2 = 0.01$). Note that reactions were faster in the non-stimming condition (591 ms) than in the stimming condition (632 ms). These effects remained significant in the group who reported stimming in their daily lives (N=14) for the effects of cue congruency ($log_{10}BF_{10} = 1.48$, $F = 11.42$, $p < 0.005$, $\eta_G^2 = 0.06$), stimming condition ($log_{10}BF_{10} = 0.42$, $F = 4.89$, $p = 0.045$, $\eta_G^2 = 0.03$), and their interaction ($log_{10}BF_{10} = -0.33$, $F = 8.71$, $p = 0.011$, $\eta_G^2 = 0.01$). For the group who did not report stimming (N=8), only the effect of cue congruency was significant ($log_{10}BF_{10} = 0.92$, $F = 40.15$, $p < 0.001$, $\eta_G^2 = 0.11$).

The EEG results from Experiment 2 are summarised in Fig 5. Panels (a,b) demonstrate that multivariate pattern analysis is able to decode cue direction, target side, and target congruency with similar timecourses to Experiment 1 (see Fig 3), though with somewhat lower accuracy overall. We then compared decoding accuracy for cue side and cue congruency between

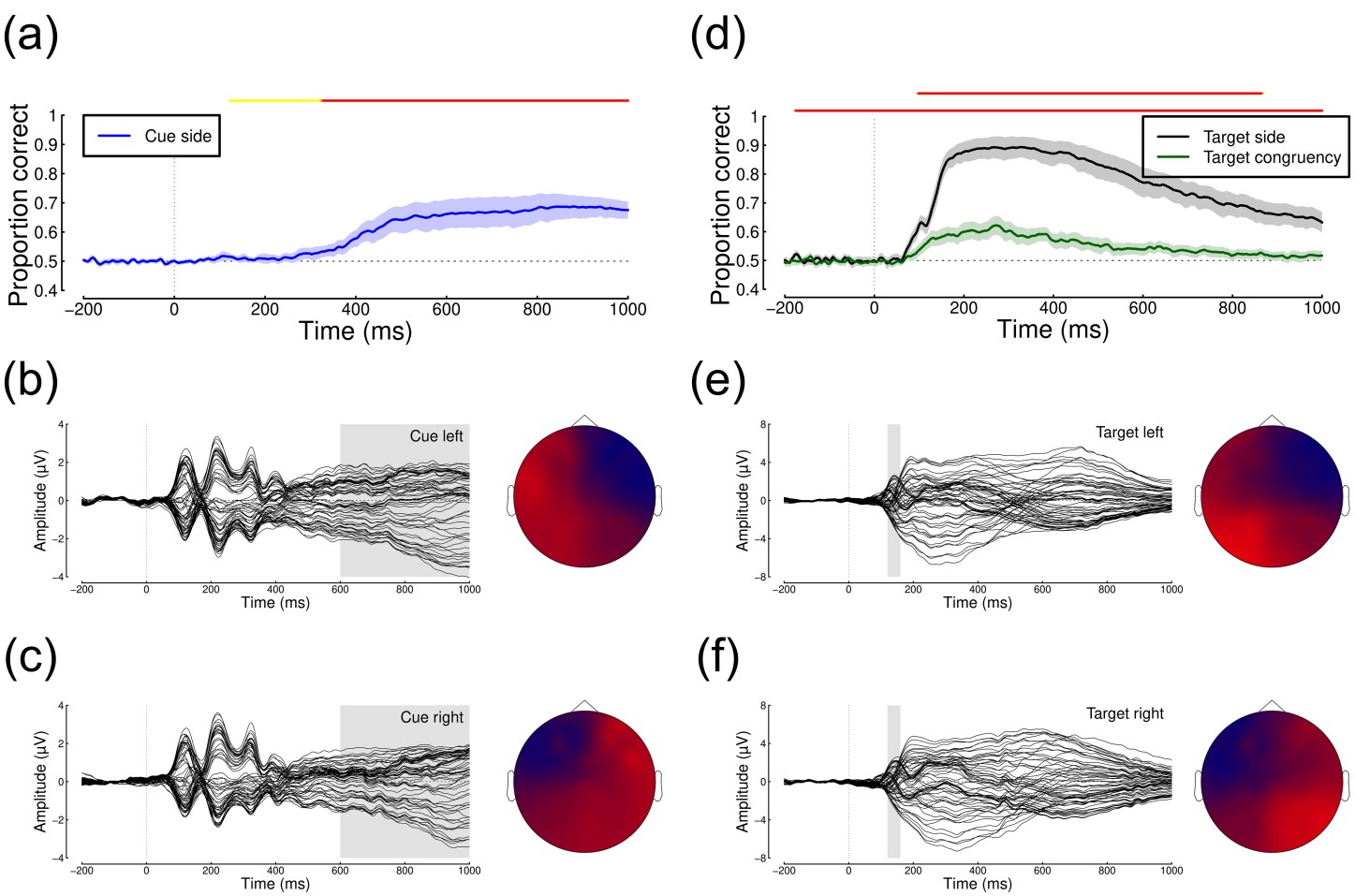

**Fig 3. Summary of EEG data from Experiment 1.** Panel (a) shows decoding accuracy of a pattern classification algorithm trained on responses to the cue stimulus (Time 0 is the cue onset). ERPs and scalp topographies are shown for cues pointing left (b) and right (c), with the shaded regions indicating the time period over which voltages were averaged to generate the scalp plots. Panel (d) shows decoding accuracy for classifying target location (black curve) or cue congruency (green curve) in response to the target (Time 0 is the target onset). ERPs and scalp topographies are shown for targets on the left (e) and right (f) side of fixation, with the shaded regions indicating the time period over which voltages were averaged to generate the scalp plots. Shaded regions in panels (a,d) indicate 95% confidence intervals, and lines above y=1 indicate Bayes factor scores above 3 (yellow >3; orange >10; red >30).

the stimming group (N=14), and the non-stimming group (N=8). Although decoding accuracy appeared to be slightly higher for the non-stimming group for cue side (Fig 5c), this difference was not convincing. Accuracy for cue congruency was very similar for the two groups (Fig 5d). Finally, we compared the deployment of spatial attention between stimming and non-stimming conditions for each of the two groups. There appeared to be a slight decrease in decoding accuracy in the non-stimming group when they were asked to stim (Fig 5f), but no differences in the stimming group (Fig 5e).

We collected eye tracking data for a subset of the participants in Experiment 2 (16/22; for the remaining participants, the eye tracker was not able to calibrate successfully). We used these data to check if participants were able to follow the task instructions and fixate centrally throughout. The results are summarised in Fig 6. The grey distributions indicate horizontal fixation positions at the time of cue onset. For most participants these are close to the central fixation (0 deg, see vertical dashed line), though there are some notable consistent offsets

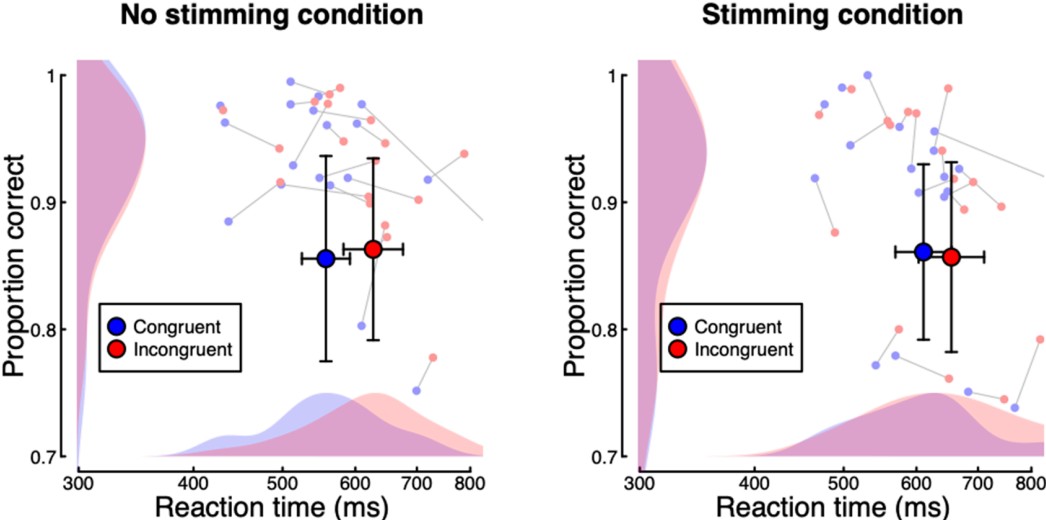

**Fig 4. Summary of reaction time and accuracy data for the cueing task in Experiment 2.** Small points show data for individual participants (N = 22), and larger points give the group means. Error bars indicate 95% confidence intervals.

that we attribute to calibration errors (e.g. P201, P223, P224). The coloured distributions indicate horizontal fixation positions at the time of target onset, following either a leftward (blue) or rightward (red) pointing cue. The majority of participants were able to follow the instructions and fixated centrally for both of these conditions, showing no systematic differences. One participant (P214) clearly made eye movements in the cued direction on most trials, and some participants (e.g. P209, P216, P218) showed evidence of sometimes making such eye movements. At a group level, there was a marginally significant difference between the mean horizontal positions for left vs right cues ($t = 2.15$, $p < 0.05$), but the overall evidence was not compelling ($log_{10}BF_{10} = 0.19$). We therefore conclude that in general participants were able to follow instructions appropriately, and that eye movements are unlikely to be responsible for the EEG effects of cueing.

## Discussion

Overall, our results from Experiment 1 support the assertion that EEG data can be used as a metric of attentional deployment [13]. The classifier was able to differentiate both cue and target side from EEG recordings in Experiment 1, indicating we have recorded a neural correlate of attentional deployment, and that it is possible for a machine learning classifier to determine the direction of spatial attention using EEG signals. Extending this method in Experiment 2 by including the stimming manipulation suggested that stimming does not have a detrimental effect on attentional deployment, with our only negative impact of stimming being on reaction times. We will now discuss these results in greater detail, with a focus on the implications for research into stimming in the context of the neurodiversity movement.

Behavioural data from Experiment 1 revealed a significant effect of congruency on both accuracy and reaction time, a typical Posner cueing effect wherein higher accuracy and shorter reaction times are associated with congruent relative to incongruent conditions [13, 14]. Experiment 1 further involved testing the classifier, which revealed that cue side was reliably decoded from 300 ms post-stimulus onset onwards, reaching an accuracy level of 69%, and target side was reliably decoded from much earlier (100 ms post-stimulus onset) and

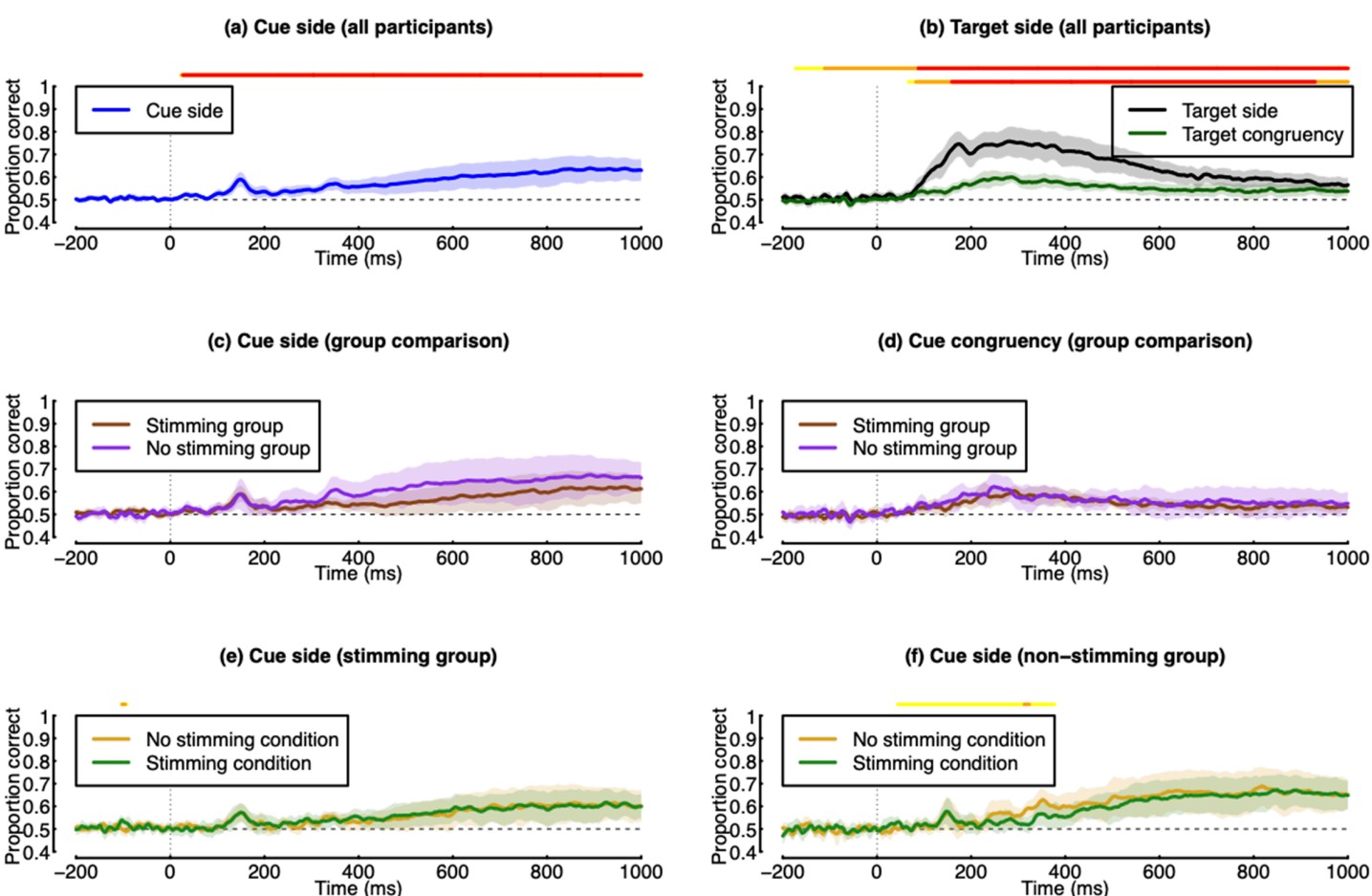

**Fig 5. Summary of EEG data from Experiment 2.** Panel (a) shows decoding accuracy of a pattern classification algorithm trained on responses to the cue stimulus (Time 0 is the cue onset) for the full sample of participants (N=22). Panel (b) shows decoding accuracy for classifying target location (black curve) or cue congruency (green curve) in response to the target (Time 0 is the target onset), again for all participants. Panels (c,d) compare decoding between the stimming (N=14) and non-stimming (N=8) participants for cue side and cue congruency. Panels (e,f) compare decoding accuracy for cue side within each group, and between the stimming and no stimming conditions. Shaded regions in all panels indicate 95% confidence intervals, and lines above y=1 indicate Bayes factor scores above 3 (yellow >3; orange >10; red >30).

reached 89% accuracy. This sufficiently validated the use of the measure in Experiment 2 and supported prior research using similar techniques [13].

Experiment 2 replicated this methodology with the addition of the stimming manipulation. Additionally, eye position data were recorded where possible, allowing us to confirm that in the majority of cases participants understood and followed the instructions, keeping their eyes fixated on the centre of the screen, and did not overtly move them following the cue. This confirms that the neural correlates we measured do not reflect eye movements, suggesting they are instead a genuine representation of the deployment of spatial attention.

Analysis of the behavioural data revealed no significant effect of either stimming or congruency on accuracy. The effect of congruency on accuracy in Posner cueing tasks is robust [13,14] and our version of the task generated this effect in Experiment 1, suggesting an external factor resulted in this difference in Experiment 2. For example, the environmental conditions under which participants completed the experiment were quite different, as data for

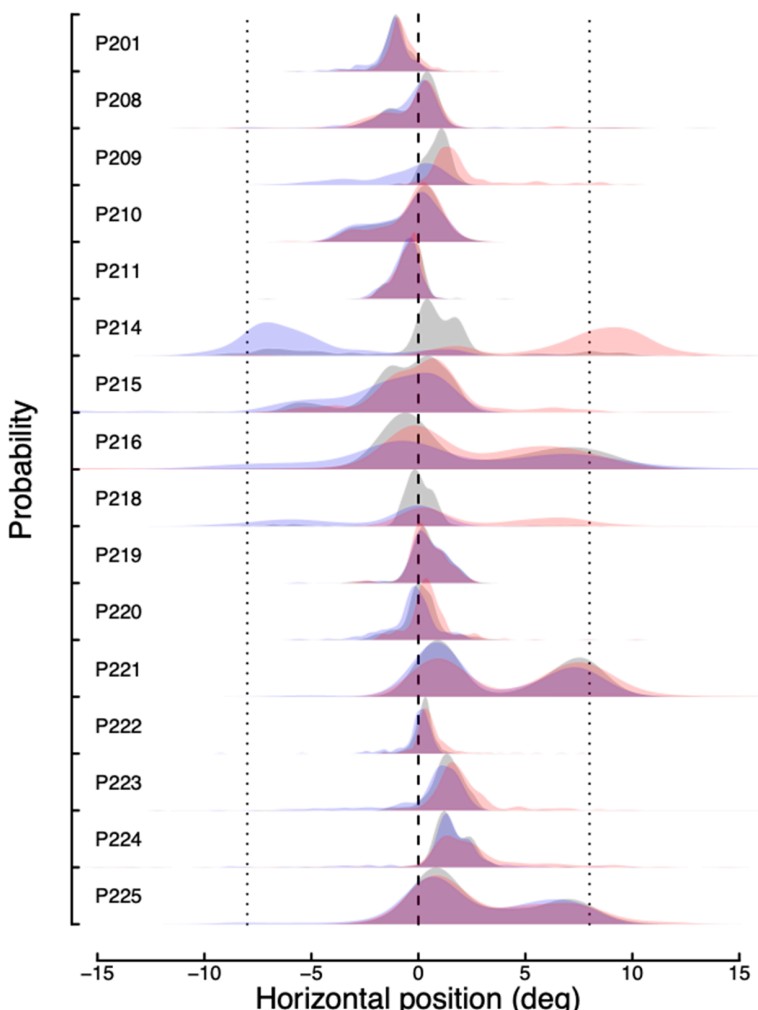

**Fig 6. Distributions of horizontal eye positions for 16 participants during Experiment 2.** Grey distributions represent eye positions at cue onset. Coloured distributions represent eye positions at target onset following a leftward pointing cue (blue) or a rightward pointing cue (red). Dotted vertical lines indicate the target locations (±8 degrees).

Experiment 1 were collected over the winter, while Experiment 2 was conducted in the summer, when the laboratory was very warm, potentially reducing participants' ability to focus on the task. Accuracy was lower overall for Experiment 2 than Experiment 1; participants in Experiment 1 averaged 96% correct for congruent trials and 93% correct for incongruent trials, while participants in Experiment 2 averaged 86% correct for both congruent trials and incongruent trials. Difficulties resulting from external factors such as temperature and general comfort may have influenced the participants' ability to attend to the experiment to the point where differences in accuracy became non-significant. Our finding that stimming had no effect on accuracy would therefore need to be validated through future replication; should it hold, it is an important one, as it suggests that stimming does not have a negative impact on individuals' performance, countering the notion that stimming behaviours are detrimental and should be targeted through interventions (e.g., [30,45]). These findings do not fully align

with self-reports from autistic individuals, however, who report that stimming benefits attention [6]; accuracy was no greater in stimming conditions than non-stimming conditions. This may result from the proportion of our participants with autism being very small, so future research should aim to further investigate this in a larger sample. It has been suggested that stimming acts are a compensatory mechanism [6] for the previously discussed attention problems associated with autism [5], so there is a possibility that positive effects of stimming are only seen in these individuals, which warrants further investigation.

Reaction times for participants in Experiment 2 were also longer than for those in Experiment 1, again suggesting the participants experienced greater difficulty when completing the task under the external conditions of the second experiment. In Experiment 1, the average reaction time was 484 ms for congruent trials and 535 ms for incongruent trials, while in Experiment 2, the average reaction time was 557 ms/611 ms for congruent trials and 627/654 ms for incongruent trials (for non-stimming/stimming conditions, respectively). However, analysis of the reaction time data did reveal a typical Posner cueing effect in regard to congruency: congruent trials were associated with shorter reaction times. Analysis also suggested a negative effect of stimming on reaction times, with reaction times being significantly slower when participants were stimming. Given the lack of impact observed on accuracy (and on EEG decoding, discussed next) it is likely this finding relates to stimming interfering with the participants' ability to complete the task (they were required to respond to stimuli with buttons on the keyboard while moving their other hand) rather than stimming being inherently distracting. Participants had to use one hand to stim while simultaneously responding to stimuli presented on the screen with the other hand - the incompatibility of the movements being performed simultaneously may therefore have caused, or contributed to, the increase in reaction times recorded. Future research into this phenomenon should modify the task to ensure stimming behaviours are compatible. Klier and Harris [46], for example, used a learning task where children learnt behaviours either compatible or incompatible with stimming and found stimming itself did not have a detrimental effect on the ability to learn. A replication of this study could use a task in which participants respond verbally, for example, to limit the effect of incompatible movements. When we further divided participants into those who reported that they stim in their everyday lives and those who reported that they did not, we found the pattern only persisted in the former group; this may be due to the much smaller size of the latter group, as for the former group the Bayes factor and effect size were smaller for the effect of stimming than the effect of congruency, suggesting the relationship between stimming and reaction time is weaker than between congruency and reaction time. In a larger group of individuals who do not typically engage in stimming, this relationship might persist. Future research should aim to recruit larger sample sizes for various neurotypes and everyday stimming behaviours. It should also aim to adapt tasks so that there is no potential for the physical motion of stimming to interfere with the physical motion of completing a task, allowing the pure effects of stimming to be investigated.

Classifier accuracy in Experiment 2 was lower overall than in Experiment 1, which was not unexpected given the smaller sample size and the fact that participants in Experiment 2 were moving, potentially introducing extra noise to the EEG signal. Within Experiment 2, classifier accuracy was largely similar between individuals who reported typically stimming and those who did not, validating comparisons between the groups based on classifier accuracy.

A decrease in classifier accuracy during stimming was observed exclusively in the group who reported that they did not typically stim. The same was not true for the typically stimming group. We consider classifier accuracy here to be a measure of the strength of an individual's ability to deploy their spatial attention, as stronger attentional deployment to the left

or right should result in a greater condition-based difference in neural activity, making it easier for the classifier to differentiate the direction of their attention. Therefore, this suggests an individual's ability to deploy their spatial attention is only reduced when they are asked to stim if they do not typically engage in the behaviour, contesting the notion that stimming is an inherently harmful behaviour. Further, it may give insight into where this concept originated; for individuals who do not typically stim, it appears to have negative consequences for their attentional abilities, which may lead them to assume that the same is true for everyone (which our evidence suggests is not the case). For individuals who typically engage in stimming behaviours, there does not appear to be an impact of stimming or not stimming on their attentional deployment, indicating that the behaviour is not detrimental; greater inclusion of neurodiverse individuals may allow for observation of a positive effect, as stimming has been proposed as a compensatory mechanism [6].

It has long been claimed that stimming behaviours interrupt attention and learning, which has been used to justify therapies aiming to reduce these behaviours (e.g., [45]). Our investigation contests this view, given our lack of evidence to suggest a negative impact of stimming on attention. The evidence often used to support this view of stimming behaviours falls foul to a host of methodological issues, such as incredibly small groups of participants [47], stopping children stimming entirely during the learning phase [48], or not attempting to prevent stimming behaviours at all [49], thereby providing no real evidence for whether or not stimming affects learning abilities, or whether reducing them would have a beneficial impact. Indeed, self-reports from autistic individuals that the behaviours benefit their attention may mean this view is leading autism interventions in entirely the wrong direction when it comes to improving the quality of life of autistic individuals. Interventions that aim to reduce stimming are worryingly widespread, given their weak research base. Behavioural approaches, particularly common in North America [50], frequently target these behaviours (e.g., [6,51]). Future research should attempt to take a similar approach to ours, investigating and allowing for the potential that stimming behaviours may have positive effects, or even no effect at all, to ensure we capture the full picture of stimming and its potential functions before encouraging interventions that target the behaviour. One factor that future research should take into consideration is that in our investigation, we directed participants to stim or refrain from stimming at certain times. It is unclear whether there is a subjective difference in stimming when it is genuinely self-generated compared to when it is externally prompted. Considering that stimming is believed to address certain internal needs, stimming when externally prompted (i.e., not in direct response to these needs) may not generate an effect, because the individual doesn't feel the need to self-regulate through stimming. This could explain the lack of a positive effect of stimming that we identified here, if only the truly self-generated form of stimming produces attentional effects, or any effects at all. A careful balance needs to be struck between controlling experimental conditions and ensuring stimming behaviours are authentic to best understand them. Future research could benefit from a more naturalistic design, in which participants complete a task while stimming when it feels natural to them, with researchers recording start and end times of stimming periods to create stimming and non-stimming conditions. Likely the ideal method of dealing with this potential conflict is for researchers across the field to conduct investigations into stimming in a number of different ways, exploiting the strengths and weaknesses of various methodologies to obtain a full picture of stimming. It would also be beneficial to ask individuals who stim directly about whether or not they believe externally prompted stimming benefits them in the same way as self-directed stimming.

Another limitation of our investigation was our use of a binary self-report relating to whether participants typically engaged in stimming or not. If one were available, it would be

beneficial to use a self-report scale designed to measure an individual's stimming behaviours. However, at present, the available scales are not appropriate for this use. The two scales we identified were the Stereotyped Behaviour Scale (SBS; [52]) and a sub-section of the Aberrant Behaviour Checklist (ABC; [53]) - both scales require an observer to rate an individual's behaviours, rather than being a self-report scale, making them unsuitable for use in this context. Further, the ABC has been criticized for subjective and unacceptable items- for example using condescending language such as 'tantrums' or describing individuals as 'whiny' [54]. To facilitate further investigations into stimming it would be beneficial to develop an appropriate self-report scale, as those currently available are not suited for this purpose.

Our investigation intended to lay the groundwork for an investigation into whether stimming behaviours in autism have the same underlying functions and mechanisms as fidgeting in ADHD. Given our limited sample size of participants with either disorder, we have not gathered sufficient evidence to suggest whether or not this is the case. However, we believe this route is a promising one for investigation under the framework of the transdiagnostic approach, especially given that there are overlaps in the benefits of stimming and fidgeting as reported by individuals with autism and ADHD. Kapp et al. [6] reported autistic individuals stated stimming benefitted their ability to focus, while Canela et al. [55] similarly reported that individuals with ADHD stated fidgeting aided in their productivity and concentration. Similarities have been identified in stimming across ASD and ADHD, with notable differences including the frequency and complexity of the movements [56]. A greater understanding of this behaviour across conditions may further our understanding of the similarities between autism and ADHD, a pressing research interest given the high comorbidity of the two conditions, with the current and lifetime prevalence rates of ADHD in the autistic population being estimated at 38.5% and 40.2% respectively [57].

Future research should additionally pursue other avenues of exploration into the potential benefits of stimming. In this case, we focused on attention, given that this is a commonly reported area in which stimming generates benefits in individuals with autism [6] and there is crossover with fidgeting in ADHD, which is also reported to benefit attention [55]. However, autistic individuals report a number of other benefits of stimming, for example emotional regulation [6,7,9,10]. In order to obtain a full picture of the mechanisms and functions of stimming, to fully understand the behaviour and thus inform interventions which aim to reduce the behaviour, future research should investigate all reported avenues in which stimming may benefit.

Given that our investigation did not uncover a positive effect of stimming on attention, it could be suggested that interventions aiming to reduce stimming behaviours are not causing harm. Often these interventions are additionally justified on the basis that children who stim may be socially excluded or bullied as a result of their stimming behaviours (e.g., [58]), and engaging in these behaviours can increase the likelihood of autistic children experiencing bullying [59], which is a serious problem. However, even if we assume stimming has no positive effect (despite self-reports from autistic individuals saying it does), efforts may be better focused on educating an autistic child's neurotypical teachers and peers on autism and stimming, cracking down on bullying, and teaching autistic children how to handle bullying. It has been demonstrated that describing and explaining autistic behaviours to children can improve their attitudes towards an autistic child [60], and a recent intervention teaching autistic children how to stand up to bullies achieved a reasonable level of success [59]. Autistic children may experience bullying as a result of a wide variety of their autistic traits [61]; surely, then, the best approach is to address the bullying, rather than encouraging autistic children to alter a wide variety of their own behaviours to avoid it. Particularly in the case of stimming, autistic adults protest against the elimination of these behaviours [6], and they have been posited

as a significant feature of autistic culture and identity [10,36,62]. There is also a growing shift in approaches to teaching autistic children, with new encouragement of accommodating and integrating stimming into the classroom environment (e.g., [63]). Even though we have been unable to demonstrate in a controlled setting that these behaviours have benefits, it is important to listen to the community we aim to help and respect their wishes. This is particularly relevant in the context of the growing movement for acceptance of neurodiversity, and the push for inclusion of autistic perspectives in research, as stimming is a sensitive topic given historical attitudes and current social stigma surrounding it.

In summary, our results do not support prior literature suggesting stimming is a behaviour with negative consequences for attention. Rather, we find that in individuals who typically engage in stimming behaviours there is no particular detriment associated with doing so (excluding delayed reaction times, which potentially are an artefact of the experimental design). Negative impacts of stimming on attentional deployment were identified in individuals who do not typically engage in stimming behaviours, which may help explain why the perception has previously been that the behaviours are detrimental. This research had a predominantly neurotypical sample, and as such it is vital for future research to include a greater sample of autistic individuals to investigate whether stimming functions in the same way for this group as it appears to for neurotypicals who stim. Furthermore, the inclusion of a greater sample of individuals with ADHD may help uncover whether the fidgeting behaviours seen in ADHD may be similar to, or the same as, autistic stimming. Future research should also aim to investigate the potential benefits of stimming in a wider range of areas, given that individuals with autism also report benefits to emotional regulation [6] in addition to attention.

## Author contributions

**Conceptualization:** Cameron Smith, Daniel H. Baker.

**Formal analysis:** Cameron Smith, Daniel H. Baker.

**Funding acquisition:** Daniel H. Baker.

**Investigation:** Cameron Smith, Daniel H. Baker.

**Supervision:** Daniel H. Baker.

**Writing – original draft:** Cameron Smith, Daniel H. Baker.

**Writing – review & editing:** Cameron Smith, Daniel H. Baker.

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
