## [Decision Letter · Decision Letter 0]

8 Jul 2025

PONE-D-25-28211Neural correlates of the deployment of spatial attention, and their modulation by repetitive movementsPLOS ONE

Dear Dr. Baker,

Thank you for submitting your manuscript to PLOS ONE. After careful consideration, we feel that it has merit but does not fully meet PLOS ONE’s publication criteria as it currently stands. Therefore, we invite you to submit a revised version of the manuscript that addresses the points raised during the review process. Please submit your revised manuscript by Aug 22 2025 11:59PM. If you will need more time than this to complete your revisions, please reply to this message or contact the journal office at plosone@plos.org. Please include the following items when submitting your revised manuscript:

We look forward to receiving your revised manuscript.

Kind regards,

Assoc. Prof. Phakkharawat Sittiprapaporn, Ph.D.

Academic Editor

PLOS ONE

Journal Requirements:

“CS was supported by a Bradshaw-Eagle Undergraduate Research Scholarship from the Applied Vision Association (https://www.theava.net/). DHB was supported by BBSRC grant BB/V007580/1. https://www.ukri.org/councils/bbsrc/

The funders played no role in the study design, data collection and analysis, decision to publish, or preparation of the manuscript.”

“5 CS was supported by a Bradshaw-Eagle Undergraduate Research Scholarship from the Applied 476 Vision Association. DHB was supported by BBSRC grant BB/V007580/1.”

“CS was supported by a Bradshaw-Eagle Undergraduate Research Scholarship from the Applied Vision Association (https://www.theava.net/). DHB was supported by BBSRC grant BB/V007580/1. https://www.ukri.org/councils/bbsrc/”

Reviewers' comments:

Reviewer's Responses to Questions

**Comments to the Author**

1. Is the manuscript technically sound, and do the data support the conclusions?

Reviewer #1: Yes

2. Has the statistical analysis been performed appropriately and rigorously? 

Reviewer #1: Yes

3. Have the authors made all data underlying the findings in their manuscript fully available?

Reviewer #1: Yes

4. Is the manuscript presented in an intelligible fashion and written in standard English?

Reviewer #1: Yes

5. Review Comments to the Author

Reviewer #1: The study relies on a binary self-report to classify participants as typically stimming or not, without using a validated scale or further behavioural detail. Authors should acknowledge this limitation explicitly in the Methods and Discussion (how this approach may mask important variability in frequency..). In addition, they have to cite and briefly describe at least one validated tool that could be used in future research to quantify stimming. Clarify in the Methods (p. 4–5) that the classification was based on a single questionnaire item and explain why this approach was chosen.

The study compares individuals who typically stim to those who do not, yet the experimental stimming task was externally imposed and not fully naturalistic. Authors should expand on the potential difference between internally motivated (self-directed) vs externally instructed stimming. Consider whether this difference may explain the absence of beneficial effects on attention. As well as authors should suggest a future study design allowing naturalistic stimming and cite any precedent for such methods if available.

Furthermore, the longer reaction times in the stimming condition may be confounded by the physical act of movement interfering with manual responses, rather than reflecting cognitive distraction. I recommend that the author should in the Discussion section be more explicit about this possible confound and its implications and propose a future adaptation of the task that avoids motor overlap.

The analysis in Experiment 2 includes only 8 participants who do not typically stim, limiting the reliability of group comparisons. Author should acknowledge the limited power of subgroup comparisons and the potential for Type II errors and suggest conducting a power analysis to estimate the required sample size for detecting group differences in decoding accuracy or reaction time under this paradigm.

Finally, while the study is well-cited, more recent citation and more clarity is needed in linking the behavioural neuroscience to the broader neurodiversity framework.

6. PLOS authors have the option to publish the peer review history of their article (what does this mean?). If published, this will include your full peer review and any attached files.

Reviewer #1: No

---

## [Decision Letter · Decision Letter 1]

3 Sep 2025

Neural correlates of the deployment of spatial attention, and their modulation by repetitive movements

PONE-D-25-28211R1

Dear Dr. Baker,

We’re pleased to inform you that your manuscript has been judged scientifically suitable for publication and will be formally accepted for publication once it meets all outstanding technical requirements.

Kind regards,

Assoc. Prof. Phakkharawat Sittiprapaporn, Ph.D.

Academic Editor

PLOS ONE

Additional Editor Comments (optional):

Reviewers' comments:

Reviewer's Responses to Questions

**Comments to the Author**

1. If the authors have adequately addressed your comments raised in a previous round of review and you feel that this manuscript is now acceptable for publication, you may indicate that here to bypass the “Comments to the Author” section, enter your conflict of interest statement in the “Confidential to Editor” section, and submit your "Accept" recommendation.

Reviewer #1: All comments have been addressed

2. Is the manuscript technically sound, and do the data support the conclusions?

Reviewer #1: Yes

3. Has the statistical analysis been performed appropriately and rigorously? 

Reviewer #1: Yes

4. Have the authors made all data underlying the findings in their manuscript fully available?

Reviewer #1: Yes

5. Is the manuscript presented in an intelligible fashion and written in standard English?

Reviewer #1: Yes

6. Review Comments to the Author

Reviewer #1: I appreciate the authors’ thoughtful and thorough revisions. All my previous concerns have been adequately addressed.

7. PLOS authors have the option to publish the peer review history of their article (what does this mean?). If published, this will include your full peer review and any attached files.

Reviewer #1: No

---

## [Editor Report · Acceptance letter]

PONE-D-25-28211R1

PLOS ONE

Dear Dr. Baker,

I'm pleased to inform you that your manuscript has been deemed suitable for publication in PLOS ONE. Congratulations! Your manuscript is now being handed over to our production team.

Kind regards,

on behalf of

Assoc. Prof. Dr. Phakkharawat Sittiprapaporn

Academic Editor

PLOS ONE